# Multi-Image Robust Alignment of Medium-Resolution Satellite Imagery

**Marco Scaioni**, **Luigi Barazzetti *** and **Marco Gianinetto**

Department of Architecture, Built Environment and Construction Engineering, Politecnico di Milano, Via Ponzio 31, 20133 Milano, Italy; marco.scaioni@polimi.it (M.S.); marco.gianinetto@polimi.it (M.G.)
* Correspondence: luigi.barazzetti@polimi.it; Tel.: +39-02-2399-8778

**Abstract:** This paper describes an automatic multi-image robust alignment (MIRA) procedure able to simultaneously co-register a time series of medium-resolution satellite images in a bundle block adjustment (BBA) fashion. Instead of the direct co-registration of each image with respect to a reference 'master' image on the basis of corresponding features, MIRA also considers those tie points that may be not be shared with the master, but they only connect the other images ('slaves') among them. In a first stage, tie points are automatically extracted by using pairwise feature-based matching based on the SURF operator. In a second stage, such extracted features are re-ordered to find corresponding tie points visible on multiple image pairs. A 'master' image is then selected with the only purpose to establish the datum of the final image alignment and to instantiate the computation of approximate registration parameters. All the available information obtained so far is fed into a least-squares BBA to estimate the unknowns, which include the registration parameters and the coordinates of tie points re-projected in the 'master' image space. The analysis of inner and outer reliability of the observations is applied to assess whether the residual blunders may be located using data snooping, and to evaluate the influence of undetected outliers on the final registration results. Three experiments with simulated datasets and one example consisting of eleven Landsat-5/TM images are reported and discussed. In the case of real data, results have been positively checked against the ones obtained by using alternative procedures (BBA with manual measurements and 'slave-to-master' registration based on automatically extracted tie points). These experiments have confirmed the correctness of the MIRA approach and have highlighted the potential of the inner control on the final quality of the solution that may come from the reliability analysis.

**Keywords:** automation; bundle block adjustment; image time-series; matching; registration; reliability analysis

## 1. Introduction

The growing availability of medium-resolution satellite images for Earth observation (EO) gives an unprecedented opportunity to monitor land cover changes and dynamic processes, up to a hyper-temporal resolution of a few days. Operating satellites such as Landsat, Disaster Monitoring Constellation, and Sentinel-2 may provide data at geometric resolution between 10 m and 30 m in terms of ground sample distance (GSD), while covering a large radiometric spectrum [1–4].

Typically, such datasets are delivered after topographic correction to remove relief displacement errors and the effect of Earth curvature [5]. These pre-processing steps result in the fact that image registration, which is a fundamental pre-requisite for any multi-image analyses, can be accomplished by using 2-D transformations [6]. The direct use of metadata information for registering image time series is not enough to carry out the precise alignment at pixel and subpixel levels, as required in many applications.

Traditional methods for *image alignment* (or *registration*) [7–9] rely on the use of corresponding features identified between a *reference image* (frequently called 'master' image) and each *generic image* ('slave' image) in the time series. This registration method is usually carried out for all the 'master-to-slave' combinations, as shown in Figure 1a. Corresponding features may be measured manually, but several procedures have been developed for their automatic extraction [10–12]. Such features are used for estimating a geometric 2D transformation to map the images to each other and to obtain pixel-to-pixel overlap after resampling. The 'master' image defines the absolute spatial datum, for example by using metadata information or *ground control points* (GCPs), which may come from in situ GNSS measurements or from existing digital maps [13].

In remote-sensing applications for change detection, the acquisition of data collected over the same geographic area but at different epochs (e.g., time of the day, season, year) may be accomplished in uneven conditions of cloud cover, illumination, viewing geometry, and/or imaging sensors [14]. Consequently, the radiometric content of a multi-temporal dataset may sharply differ from one image to another [15], even if the sensed areas overlap. In such a case, the image alignment of long time series may become an involved task. This may also result in the fact that all 'slave' images may not share enough corresponding features with the 'master' image, resulting in need of concatenating additional 'master' images with consequent error propagation.

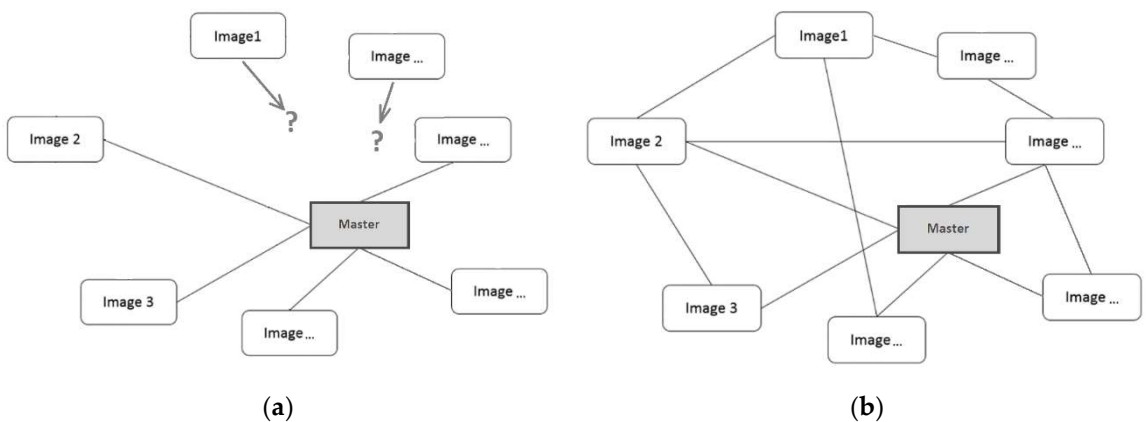

(**a**)　　　　　　　　　　　　　　　　　　　　　(**b**)

**Figure 1.** Traditional 'master-to-slave' (**a**) versus multi-image alignment (**b**). Lines represent the existence of enough corresponding features shared between two images to enable the computation of the registration parameters.

Starting from Orun and Natarajan [16], an alternative approach for the registration of satellite images on the basis of *bundle block adjustment* (BBA) was introduced. BBA is commonly applied in photogrammetry and computer vision for image orientation [17]. The basic concept is to use not only corresponding features for estimating the 'master-to-slave' registration parameters, but also to introduce corresponding points shared between the 'slave' images, see Figure 1b. The coordinates of these correspondencies, which are usually addressed to as *tie points* (TPs), are used to instantiate a system of redundant equations to be solved within a least-squares (LS) framework for the determination of the unknown registration parameters and the coordinates of TPs in a given geodetic *datum*. This can be defined by introducing enough GCPs or using some inner constraints [18]. In such a way, also those 'slave' images without corresponding features that are directly shared with the 'master' may be registered, limiting error propagation. In addition, since TPs may be observed on more than two images and used to write multiple equations in the BBA formulation, the *inner reliability* of the observations increases and the procedure may gain robustness against gross errors. Moreover, some additional images could be added into the data processing workflow to improve the network geometry and to benefit to the alignment.

The application of BBA approach to the registration of satellite data has been already proposed in the technical literature. Toutin [19] developed a solution for the BBA of Landsat 7 ETM+ based on a

3D analytical geometric model for multi-sensor images, including orbital constraints. Different sets of 3D GCPs integrated to TPs with only known elevations were tested. Results obtained from BBA were similar to the ones from single 'image-to-GCP' alignment, but with a significant reduction of GCP number. The same approach was then extended to deal with high-resolution Ikonos data [20] and to multi-sensor fusion [21,22]. At the same time, Grodecki and Dial [23] investigated a similar technique but based on *rational polynomial coefficients* (RPCs). Since this model can deal with a large variety of sensors, it applies to any imaging systems with a narrow field-of-view, a calibrated stable interior orientation, and an accurate a priori exterior orientation. The same research direction was continued in Fraser and Hanley [24] and Rottensteiner [25], where a bias-compensated BBA based on RPCs was demonstrated to provide subpixel accuracy notwithstanding the minimum ground control. High-resolution Ikonos, QuickBird, and ALOS data were used in this study. As far as new sensors have been launched, new approaches for BBA of high-resolution satellite images were introduced, as in the case of Chinese ZY3 data [26].

The studies mentioned above mainly focus on the analytical aspects of BBA and the evaluation of the obtainable accuracy when using the proposed methods. In Barazzetti [27] the focus was given on the automatic extraction of TPs for the registration of medium-resolution satellite image sequences, instead of using manual interactive measurements. These TPs are obtained on the basis of robust *feature-based matching* (FBM) techniques (see Section 2.1), and then input in the BBA for the simultaneous computation of all image registration parameters. Of course, in the case of poor image texture, the automatic extraction of TPs may easily fail. This case frequently happens, for instance, when a significant portion of the image depicts a water body. On the other hand, this problem does not depend on the method used for the measurement of TPs, since, in the case of poor image texture, the interactive approach may also result in severe problems.

The automatic extraction of TPs is a fundamental step towards the complete automation of the registration process, which represents a task of high relevance when dealing with large datasets. An important objective in automatic procedures is to minimize the influence of possible residual measurement errors. Even though FBM may limit the number of blunders, also a small fraction of them in the set of observations to be processed within the BBA might lead to significant biases in the final image registration parameters. The methods usually adopted to reject outliers in BBA are more efficient when data redundancy is large. This property may be obtained on one side by extracting multiple connections between images. On the other hand, the application of the *inner* and *outer reliability* concepts, widely popular in geodesy and photogrammetry, also allows evaluating which is the risk to have registration errors larger than prefixed thresholds. Thus, the focus in this paper is given to develop the concept presented in Barazzetti [27] to be integrated into a full, robust procedure (multi-image robust alignment) for the automatic registration of medium-resolution satellite images, with special emphasis on the reliability analysis (see Section 2). In Section 3 an application to a dataset of Landsat-5/TM images is presented. After a discussion on the experimental results in Section 4, Section 5 draws some conclusions and addresses future work.

## 2. Methods

### 2.1. Overview

The multi-image robust alignment (MIRA) procedure consists in a multi-step process, as shown in Figure 2. The core is a preliminary subdivision of the original dataset made up of n images into all $n(n - 1)/2$ possible image-pair combinations. Image correspondences are looked for independently in each image pair by using FBM, as described in Section 2.2. In a second stage, all the extracted features are reordered to find multiple TPs (Section 2.3). A reference 'master' image is selected with the only purpose to set up the spatial *datum*. This selection is based on the analysis of extracted connections between images (see Section 2.4). The TP set is used to jointly estimate all transformations' parameters

within a BBA (see Section 2.5). One crucial aspect of this procedure is the analysis of observations' inner reliability, which will be the subject of Section 2.6.

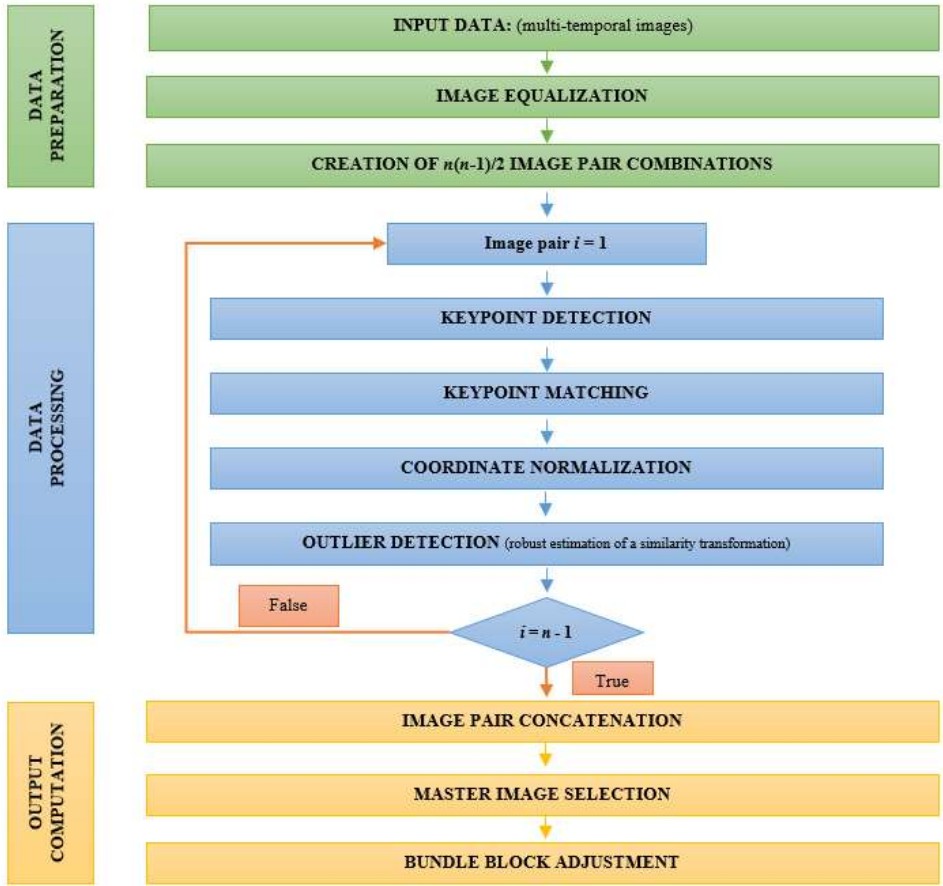

**Figure 2.** Flowchart of the MIRA method.

## 2.2. Analysis of Individual Image Pairs

After the publication of SIFT [28], a new category of *SIFT-like* algorithms has started to be successfully applied in FBM procedures for the registration of images in close-range photogrammetry and computer vision (see Wu [29]). Thanks to a multi-resolution analysis, SIFT-like algorithms are able to extract distinctive features in the images with a high degree of robustness against shift, scale, and rotation. Each feature is also assigned a *descriptor* (for example, a vector of 128 elements that can be characterized using its norm) that may be exploited for matching features in different images. SIFT-like algorithms have replaced *correlation-based* techniques that were previously used in FBM. These were based on the similarity between radiometric values and consequently were less robust against geometric and radiometric changes [11].

Here, the SURF operator has been specifically used [30]. SURF has a lower computational time than other SIFT-like algorithms [31,32] and is prone to extract a large number of *key-points* (i.e., points that are candidates to become TPs) that may be observed in more than a single image pair (*manifold* features), as proved in Barazzetti [33]. This capability is very important to obtain a redundant dataset of TPs, which is the primary purpose of MIRA.

Corresponding features are obtained by exhaustively comparing the SURF descriptors (in vectors $D_m$ and $D_n$) of all key-points extracted on the image pair to register. No preliminary information about the spatial location in the images is used for registration. Then the Euclidean distance ($d_{mn}$) between descriptors of points in different images is used as metrics for FBM:

$$d_{mn} = ||D_m - D_n||,$$ 

(1)

The following procedure is applied to find a set of homologous points:

1. Distances $d_{mn}$ are worked out between all combinations of key-points in the image pair;
2. Distances $d_{mn}$ are ranked from the shortest $d^1_{mn}$ to the largest $d^p_{mn}$;
3. The couple of key-points corresponding to the smallest distance $d^1_{mn}$ are assumed to be matched;
4. A *ratio test* [34] (first distance $d^1_{mn}$/second distance $d^2_{mn}$) is applied to scrutinize distinctive matches;
5. If the ratio test has passed, both key-points are assigned as corresponding points and are removed from the list of potential matches; otherwise, the points are kept available to be considered at a later stage; and
6. The analysis moves on to consider the following distance $d^2_{mn}$ in the rank up to the completion of the list.

When SURF retrieves a sufficient number of image correspondences, some mismatches may often still be present. To remove such outliers, the procedure makes use of the robust estimation of a 2D similarity transformation between both sets of extracted features in the images. In Barazzetti [27] more details can be found about the implementation of this step, which is based on the use of the popular high-breakdown point estimator RANSAC [35].

The similarity transformation may not be the best model to fit real data, depending on the sensor and the ground topography [13]. On the other hand, at this stage, the aim is to remove large blunders, which may account for tens of pixels. The recourse to a similarity transformation may help cope with this problem, while the rejection of smaller measurement errors is afforded afterward by using more refined models (see Table 1) that may be selected during the successive multi-image adjustment. For this reason, a relatively large threshold (e.g., 2–3 pixels) is selected for discriminating outliers with RANSAC.

The geometric distribution of corresponding points is analyzed to seek for weak configurations during RANSAC estimation. The covariance matrix $\mathbf{C}_{xx}$ of the registration parameters is compared against a *criterion matrix* $\mathbf{H}$ constructed by assuming an optimal point distribution. The analysis is based on the maximum eigenvalue $\lambda_{max}$ of matrix $\mathbf{K}$, which is computed as follows:

$$\mathbf{K} = \left(\frac{n_C}{n_H}\right)\mathbf{H}^{\frac{1}{2}}\mathbf{C}_{XX}\mathbf{H}^{\frac{1}{2}} \tag{2}$$

**Table 1.** Geometric registration models encompassed as subcases of the general planar transformation implemented in MIRA during multi-image adjustment.

| Mapping Model | # Unknowns ($n_p$) | Equations | $N_{min}$ | Geometric Deformations |
|---|---|---|---|---|
| Similarity | 4 | $x_i = a_0 + x_j m \cos\alpha - y_j m \sin\alpha$<br>$y_i = b_0 + x_j m \sin\alpha + y_j m \cos\alpha$ | 12 | 2D shifts, rotation, scaling |
| 1st Degree Polynomial (Affine) $u = 1$ | 6 | $x_i = a_{00} + a_{10}x_j + a_{11}y_j$<br>$y_i = b_{00} + b_{10}x_j + b_{11}y_j$ | 18 | 2D shifts, rotation, shear, scaling along both axes |
| 2nd Degree Polynomial ($u = 2$) | 12 | $x_i = a_{00} + a_{10}x_j + a_{11}y_j +$<br>$+ a_{20}x_j^2 + a_{21}x_jy_j + a_{22}y_j^2$<br>$y_i = b_{00} + b_{10}x_j + b_{11}y_j +$<br>$+ b_{20}x_j^2 + b_{21}x_jy_j + b_{22}y_j^2$ | 36 | 2D shifts, rotation, scaling along both axes, torsion, convexity along both axes |
| 3rd Degree Polynomial ($u = 3$) | 20 | $x_i = a_{00} + a_{10}x_j + a_{11}y_j + a_{20}x_j^2 + a_{21}x_jy_j +$<br>$+ a_{22}y_j^2 + a_{30}x_j^3 + a_{31}x_j^2y_j + a_{32}x_jy_j^2 + a_{33}y_j^3$<br>$y_i = b_{00} + b_{10}x_j + b_{11}y_j + b_{20}x_j^2 + b_{21}x_jy_j +$<br>$+ b_{22}y_j^2 + b_{30}x_j^3 + b_{31}x_j^2y_j + b_{32}x_jy_j^2 + b_{33}y_j^3$ | 60 | 2D shifts, rotation, scaling along both axes, torsion, and convexity along both axes, other deformations without geometric interpretation |

Here the approach proposed in Förstner and Wrobel [36] has been modified to account for the different number of points in the real ($n_c$) and the ideal case ($n_h$), see Syrris [15]. If $\lambda_{max} \leq 1$ the real configuration is better than the ideal one. In such a case, the solution is accepted, and the RANSAC

estimate is terminated. If the condition $\lambda_{max} \leq 1$ is never verified, the solution corresponding to the minimum eigenvalue, but less than a safety threshold ($\lambda_{th}$ = 4) is selected.

After the completion of RANSAC cycle, the set of inliers is rejected if the number of corresponding points is below $N_{min}$ = 12, otherwise they are used for estimating the four parameters of the similarity transformation on the basis of LS regression [37]. The threshold $N_{min}$ corresponds to six times the minimum dataset to compute a similarity transformation (i.e., two points), which is twice the safety value (three) frequently adopted in geodesy to guarantee a sufficient global redundancy in LS regressions. A statistical testing procedure based on *data snooping* [38] is applied to remove the possible remaining tiny fraction of outliers (<5%).

The quality assessment of the solution is based on checking the theoretical accuracy of shift parameters ($\hat{\sigma}_c$ and $\hat{\sigma}_r$ along columns and rows, respectively) that may be extracted from the estimated covariance matrix $\mathbf{C}_{xx}$. These are related to the standard deviation of an observation with unit weight ($\hat{\sigma}_0$) estimated after LS regression, and to the number of extracted inliers (F):

$$\hat{\sigma}_c = \hat{\sigma}_r = \frac{1}{\sqrt{F}}\hat{\sigma}_0. \tag{3}$$

A threshold on $\hat{\sigma}_c$ and $\hat{\sigma}_r$ is introduced to check weak configurations. In fact, results worse than $\hat{\sigma}_c = \hat{\sigma}_r = \pm 3$ pixels for shift parameters are mainly caused by incorrect matches and should be removed from the datasets.

It should be also mentioned that the FBM process is highly prone to be parallelized. On one side the extraction of key-points with the SURF operator may be independently carried out in each image. On the other, the pairwise FBM procedure may be applied to each considered image pair disregarding others.

*2.3. Derivation of Multiple Tie Points*

As expected from the multiple overlaps, it is likely that some corresponding features are 'visible' on more than a single image pair. Indeed, as already addressed in Section 2.2, an essential characteristic of SURF operator is the capability of finding the same feature in multiple images under different geometric and radiometric conditions. A comparison of the numerical value of the extracted pixel coordinates of corresponding points obtained from the image-to-image matching stage may provide a regular structure of multiple (or *manifold*) tie points (TPs), i.e., TPs that may be measured on more than two images. Such points improve the inner reliability of the observations and make the registration process less sensitive to residual measurement errors [39], as it will be illustrated in Section 2.6.

*2.4. Selection of the 'Master' Image*

The selection of the 'master' image is based on the distribution of corresponding points between the images. A *connectivity graph* is drawn to show the relationship between the images and to highlight possible weak connections in the network. An example of connectivity graph related to the example presented in Section 3 is displayed in Figure 3 using a matrix representation, where a cross indicates the presence of sufficient TPs for co-registering the pair of images corresponding to the intersecting row and column. While a rapid look at the *connectivity matrix* is enough to check out its consistency in the case of visual interpretation, a test is implemented to afford this task automatically. Once the consistency of the connectivity graph is assessed, the 'master' image is chosen on the basis of the following criteria:

1. to maximize the number of images that share enough TPs with the 'master', i.e., at least $N_{min}$; and
2. the 'master' image should be preferably in a central position in the time series t = 1, 2, 3, . . . , n. Indeed, a time series covers the same spatial region within a time span and, consequently, the temporal factor is more important than the spatial factor for selecting the 'master' image.

The connectivity graph is also useful to work out the approximate values for any parameters to be estimated in the global multi-image BBA, see Section 2.5. The adopted procedure is similar to the scheme followed in the 'structure-from-motion' technique [40] for image orientation in close-range photogrammetry. By looking at the connectivity graph, all 'slave' images that are directly linked to the 'master' image may be approximately registered in pairwise, independent manner. In this way, any images in this first group of registered images can be used as new 'master' images to register other 'slaves' that would share sufficient corresponding points with one of them. By using such approximate registration parameters it is then possible to re-project the coordinates of any points on the space of the main 'master' image.

| TM1 (0.45-0.52 μm) | | | | | | | | | | | | |
|---|---|---|---|---|---|---|---|---|---|---|---|---|
| IID | 1 | 2 | 3 | 4 | 5 | 6 | 7 | 8 | 9 | 10 | 11 | IID |
| 1 | ■ | X | X | X | X | X | X | X | X | X | X | 1 |
| 2 | X | ■ | X | X | X | X | X | X | X | X | X | 2 |
| 3 | X | X | ■ | X | X | X | X | X | O | X | X | 3 |
| 4 | X | X | X | ■ | X | X | O | X | X | X | X | 4 |
| 5 | X | X | X | X | ■ | X | X | X | X | X | X | 5 |
| 6 | X | X | X | X | X | ■ | X | X | X | X | X | 6 |
| 7 | X | X | X | X | X | X | ■ | X | X | X | X | 7 |
| 8 | X | X | X | X | X | X | X | ■ | X | X | X | 8 |
| 9 | X | X | X | X | X | X | X | X | ■ | X | X | 9 |
| 10 | X | X | X | X | X | X | X | X | X | ■ | X | 10 |
| 11 | X | X | X | X | X | X | X | X | X | X | ■ | 11 |
| IID | 1 | 2 | 3 | 4 | 5 | 6 | 7 | 8 | 9 | 10 | 11 | IID |
| TM2 (0.52-0.60 μm) | | | | | | | | | | | | |

| TM3 (0.63-0.69 μm) | | | | | | | | | | | | |
|---|---|---|---|---|---|---|---|---|---|---|---|---|
| IID | 1 | 2 | 3 | 4 | 5 | 6 | 7 | 8 | 9 | 10 | 11 | IID |
| 1 | ■ | X | X | X | X | X | X | X | X | X | X | 1 |
| 2 | X | ■ | X | X | X | X | X | X | X | X | X | 2 |
| 3 | O | O | ■ | X | X | X | X | X | X | X | X | 3 |
| 4 | O | O | X | ■ | X | X | X | X | X | X | O | 4 |
| 5 | O | O | O | O | ■ | X | X | X | X | X | X | 5 |
| 6 | O | O | O | O | X | ■ | X | X | X | X | X | 6 |
| 7 | O | O | O | O | O | O | ■ | X | X | X | X | 7 |
| 8 | O | O | O | O | O | O | O | ■ | X | X | X | 8 |
| 9 | O | O | O | O | O | O | O | O | ■ | X | X | 9 |
| 10 | O | O | X | O | O | O | O | X | X | ■ | X | 10 |
| 11 | O | O | X | X | O | O | O | X | X | X | ■ | 11 |
| IID | 1 | 2 | 3 | 4 | 5 | 6 | 7 | 8 | 9 | 10 | 11 | IID |
| TM4 (0.76-0.90 μm) | | | | | | | | | | | | |

| TM5 (1.55-1.75 μm) | | | | | | | | | | | | |
|---|---|---|---|---|---|---|---|---|---|---|---|---|
| IID | 1 | 2 | 3 | 4 | 5 | 6 | 7 | 8 | 9 | 10 | 11 | IID |
| 1 | ■ | X | X | X | X | X | X | X | X | X | X | 1 |
| 2 | X | ■ | X | X | X | X | X | X | X | X | X | 2 |
| 3 | X | X | ■ | X | X | X | X | X | X | X | X | 3 |
| 4 | X | X | X | ■ | X | X | X | X | X | X | X | 4 |
| 5 | O | X | X | X | ■ | X | O | X | X | X | X | 5 |
| 6 | X | X | X | X | X | ■ | X | X | X | X | X | 6 |
| 7 | X | X | X | X | O | X | ■ | X | X | X | X | 7 |
| 8 | X | O | X | X | X | X | X | ■ | X | X | X | 8 |
| 9 | X | X | X | X | X | X | X | X | ■ | X | X | 9 |
| 10 | X | X | X | X | X | X | X | X | X | ■ | X | 10 |
| 11 | X | X | X | X | X | X | X | X | X | X | ■ | 11 |
| IID | 1 | 2 | 3 | 4 | 5 | 6 | 7 | 8 | 9 | 10 | 11 | IID |
| TM7 (2.08-2.35 μm) | | | | | | | | | | | | |

**Figure 3.** Connectivity matrices for different spectral bands of 'Multan' dataset (see Section 3). In the upper and lower triangles of each matrix, a single spectral band is represented. The content of the cell at the intersection of two image identifiers (IIDs) shows the presence of sufficient ('X') or insufficient ('O') TPs between those images.

## 2.5. Analysis of Multiple Images

Since the MIRA method provides corresponding TPs not only for 'master-to-slave' combinations but also for 'slave-to-slave' pairs, all transformation parameters can be concurrently estimated.

A generic 2D polynomial transformation of degree p between two images is implemented at this stage. Given a feature k, the transformation of its coordinates between images i and j is given by:

$$\begin{cases} x_{jk} = \sum_{u=0}^{p} \sum_{v=0}^{u} a_{uvj} x_{ik}^{u-v} y_{ik}^{v} \\ y_{jk} = \sum_{u=0}^{p} \sum_{v=0}^{u} b_{uvj} x_{ik}^{u-v} y_{ik}^{v} \end{cases} . \tag{4}$$

The number ($n_p$) of transformation parameters ($a_{uvj}$, $b_{uvj}$) depends on the selected geometric model (see Table 1).

The implementation of Equation (4) in the BBA has been revised with respect to the previous version published in Barazzetti [27] to define a more rigorous stochastic model. All transformations in the global adjustment are written between the 2D image space of the generic 'slave' j and the 2D image space of the 'master' (M). The *functional model* of the observation equations based on Equation (4) then becomes:

$$\begin{cases} v_{xjk} + x_{jk} = \sum_{u=0}^{p} \sum_{v=0}^{u} \underline{a}_{uvj} \underline{x}_{Mk}^{u-v} \underline{y}_{Mk}^{v} \\ v_{yjk} + y_{jk} = \sum_{u=0}^{p} \sum_{v=0}^{u} \underline{b}_{uvj} \underline{x}_{Mk}^{u-v} \underline{y}_{Mk}^{v} \end{cases} , \tag{5}$$

where $\underline{x}_{Mk}$ and $\underline{y}_{Mk}$ are the image coordinates of tie point k projected in the image space of the 'master' image, while $v_{xjk}$ and $v_{yjk}$ are residuals. Underlined parameters in Equation (5) are considered as unknowns, including TP coordinates re-projected on the 'master' image. This formulation is stochastically corrected, since the observed quantities (with errors) are bounded on the left side and unknowns are on the right side. Consequently, the observed coordinates ($x_{jk}, y_{jk}$) of any TPs may be properly weighted if their measurement precision is uneven. Equation (5) is not linear in the parameters and requires linearization around approximate values (see Section 2.4). Consequently, the unknown parameters in Equation (5) are replaced by corrections $\underline{da}_{uvj}$, $\underline{db}_{uvj}$, $\underline{dx}_{MK}$, and $\underline{dy}_{Mk}$.

Since it is not possible to predict the accuracy of individual features extracted by SURF, the observations are assigned unit weights. In the case a TP is cast into Equation (5), its coordinates on the 'master' image will have to be fixed to the measured value. A set of additional pseudo-observation equations is introduced to keep corresponding corrections constrained to zero. Additional constraint equations may also be included in the functional model to enforce conditions between parameters, for example, when a first-degree polynomial has to be reduced to a similarity transformation.

A TPs found on $n_{im}$ image (see Section 2.3) provides $2n_{im}$ observation equations. The total number of unknowns depends on the number of images (n), the parameters of the adopted geometric transformation ($n_p$), the coordinates of 'slave-to-slave' matches re-projected onto the 'master' image ($2n_{rep}$), and the number of 2D points (2u) used as a reference (GCPs) on the 'master' image.

The system of observation, pseudo-observation and constrain equations can be rewritten in compact form as follows:

$$\begin{cases} v + y = \mathbf{A}_1 dx_1 + \mathbf{A}_2 dx_2 + c \\ \quad -w_1 = \mathbf{I} dx_2 \\ \quad -w_2 = \mathbf{D} dx_1 \end{cases} , \tag{6}$$

where:

$dx_1$:  vector of corrections to the transformation parameters between any 'slaves' and the 'master' image;

$dx_2$:  vector of corrections to point coordinates on the 'master' image;

$\mathbf{A}_1, \mathbf{A}_2$:  coefficient (or *design*) matrices of parameter vectors $dx_1$ and $dx_2$;

$y$:  vector of measured coordinates of TPs;

$c$:  vector of constants in linearized observation equations;

$v$, $w_1$, $w_2$: vectors of residuals; and

**D**:　　　　coefficient matrix of additional constraint equations, if any.

After casting all unknown parameters into a single unknown vector $dx = [dx_1 \ dx_2]^T$ the *design* matrix can be redefined as follows:

$$\mathbf{A} = \begin{bmatrix} \mathbf{A}_1 & \mathbf{A}_2 \\ \mathbf{0} & \mathbf{I} \\ \mathbf{D} & \mathbf{0} \end{bmatrix}, \tag{7}$$

The system of linearized equations may be solved to estimate the vector of unknowns $\hat{x} = \mathbf{N}^{-1}b$, where $\mathbf{N} = \mathbf{A}^T\mathbf{W}\mathbf{A}$ is the *normal* matrix and $b = \mathbf{A}^T\mathbf{W}Y$, while $Y = [c{-}y \ 0 \ 0]^T$ is the *constant vector* and $\mathbf{W}$ the weight matrix. The theoretical accuracy of the solution may be derived from the estimate of the covariance matrix $\mathbf{C}_{xx} = \hat{\sigma}_0^2\mathbf{N}^{-1}$, where $\hat{\sigma}_0^2$ is the estimated variance of unit weight observations (or sigma naught).

Data snooping is applied again on the residuals after BBA to remove small errors. The effectiveness of this procedure is directly related to the evaluation of reliability that is described in the following section.

## 2.6. Analysis of the Reliability of Multi-Image BBA

With the term *reliability* the chance to identify a gross error in the observations (*inner reliability*) and the estimated parameters (*outer reliability*) is referred to. While the readers may find the theoretical background about reliability analysis in Förstner [41] and Kraus [42], the basic concepts are briefly reviewed in the following of this section.

It is well known that a gross error in the $i$th observation may be reflected only to a limited extent into the corresponding residual $v_i$ after BBA. Thus, the largest residual does not necessarily correspond to a gross error. Moreover, outliers and random errors mask one another, so that the localization of gross errors may be difficult. In the case no gross errors are in the observations, the probability distribution of each residual is supposed to be Gaussian as $N(0, \hat{\sigma}_{vi}^2)$. The estimated variance of residual $v_i$ may be derived from the estimated covariance matrix of residuals:

$$\mathbf{C}_{vv} = \hat{\sigma}_0^2\left(\mathbf{W}^{-1} - \mathbf{A}\mathbf{N}^{-1}\mathbf{A}^T\right). \tag{8}$$

The *data snooping* [38] technique implemented for scrutinizing gross errors is based on the analysis of standardized residuals $z_{i \,=\,} v_i/\sigma_{vi}$, which are expected to be distributed as a standard Gaussian probability density function $N(0,1)$. In the hypothesis that each $z_i$ should follow the distribution $N(0,1)$, an upper threshold $k_\alpha$ on the absolute size of acceptable standardized residuals is then fixed to filter out possible outliers. An observation is accepted only if the corresponding standardized residual $|z_i| \leq k_\alpha$, where $k_\alpha$ is directly related to a given *risk probability* ($\alpha$) that $|z_i| > k_\alpha$ (see Figure 4). The rejection might occur in two different cases: (1) the residual is supposed not to follow the distribution $N(0,1)$ because the related observation is a gross error; or (2) a rare event belonging to the distribution $N(0,1)$ but with low risk probability $\alpha$ has happened. In such a case, an inlier would be erroneously discarded (type I error).

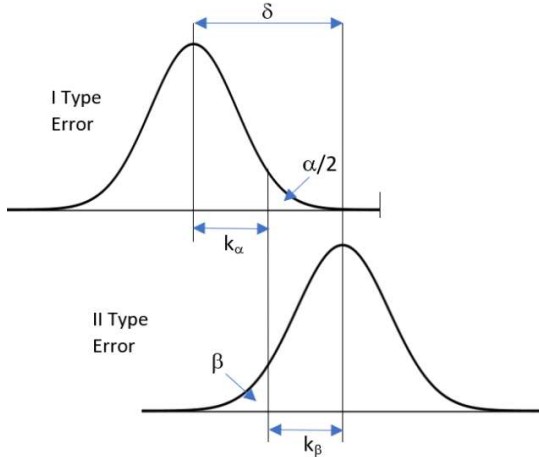

**Figure 4.** Graphical visualization of type I and II errors.

The test adopted to scrutinize the residuals may also fail in the presence of an outlier that does not follow the standard Gaussian distribution, but has a biased expectation $E(z_i) = \delta$. As shown in Figure 4, a small outlier may not be detected because it features a corresponding standardized residual $z_i$ that is smaller than the acceptance threshold $k_\alpha$. The probability of accepting an outlier (type II error) is given by $\beta$, which is related either to the risk probability $\alpha$ and to the bias $\delta$. This last parameter is commonly referred to as the *non-centrality parameter* ($\delta$) and gives in standardized coordinates the size of the minimum measurement error that may be detected using data snooping. Once the *risk probability* ($\alpha$) and the power of the test ($\beta$) have been selected, the *non-centrality parameter* ($\delta$) can be worked out. A typical set of parameters that has been also adopted in the experiments reported in this paper is: $\alpha = 1\%$, $\beta = 93\%$, $k_\alpha = 2.56$, and $\delta = 4$.

The *inner reliability* ($E(\Delta l_a)_{rel}$) of the observation $i$th is defined as the size of the minimum detectable error, according to selected parameters $\alpha$ and $\beta$. Its corresponding value can be computed using the expression:

$$E(\Delta l_a)_{rel} = \delta \sigma_{wi} / \sqrt{r_i}, \qquad (9)$$

where $\sigma_{wi}$ is the expected precision of the $i$th observation and $r_i$ is its *local redundancy*, which may be obtained from the $i$th element of the main diagonal of the *redundancy matrix* **R**:

$$\mathbf{R} = \left( \mathbf{W}^{-1} - \mathbf{A}\mathbf{N}^{-1}\mathbf{A}^{\mathrm{T}} \right) \mathbf{W}. \qquad (10)$$

Equation (9) shows that the test on standardized residuals may leave undetected a gross error whose size is $\delta_i$ times the precision of the observation $i$th, given prefixed values for $\alpha$ and $\beta$. On the other hand, the local redundancy $r_i$ may have a mitigating effect on the inner reliability. Higher values of $r_i$ lead to a smaller size for the detectable errors in corresponding observations. In a BBA, a manifold observation results in a high $r_i$ and consequently in a better inner reliability.

The final step is to evaluate the effect of an undetected gross error on the estimated parameters. This may be computed through the corresponding *outer reliability*:

$$E(\Delta |x_a|)_{rel} = \left( \mathbf{A}^{\mathrm{T}}\mathbf{W}\mathbf{A} \right)^{-1} \mathbf{A}^{\mathrm{T}}\mathbf{W}\Delta l_a, \qquad (11)$$

where vector $\Delta l_a$ reports the value of inner reliability $E(\Delta l_a)_{rel}$ on the line $i$th and zero elsewhere.

To demonstrate how the inner and outer reliabilities may be obtained in the case of a BBA adopted to register a satellite time series implementing a 2D similarity transformation model, the following simulated examples are proposed (see also Scaioni [43]). In Table 2 the range of inner reliabilities and their average values are shown for three different cases. Additionally, the outer reliability corresponding to the average inner reliability is reported in the same table.

In Case 1, two images have been registered by using 16 corresponding features. It may be seen that the limit for the minimum detectable error corresponds to a bias of 0.27 pixels in the estimated shift in the same direction. In Case 2, a total number of three 'slave' images have been included, in addition to the 'master'. Any 'slaves' share 16 TPs with the 'master'. 'Slaves' also share nine TPs among them. As may be seen from Table 2, the inner reliability of TPs shared with the 'master' is only slightly better than the one in Case 1. Looking at TPs between 'slaves', the minimum detectable error is larger with respect to the previous group of TPs. However, when looking at the corresponding outer reliability, the maximum detectable errors lead to biases in the estimated parameters that are significantly smaller than in Case 1.

In Case 3, the dataset adopted in Case 2 has been integrated by three additional images, leading to a total number of six 'slaves.' New images are not directly connected to the 'master', but they share 16 TPs among them.

The result regarding the inner reliability shows a further improvement concerning both Cases 1 and 2. Similarly, TPs have a higher threshold for non-detectable errors when they are shared among 'slave' images only. In Table 2 the inner reliabilities have been separately computed for TPs shared between three (as in Case 2) and six 'slave' images, respectively. As expected, values of inner reliability are lower in the case of points visible on six images (Subset 'SS6') than in the case of points visible on three images only (Subset 'SS3'). It is also interesting to observe how the outer reliability has improved in Case 3 concerning the previous cases with less redundant observations. In particular, the effect of maximum non-detectable gross errors in TPs is quite low (less than 1/10 pixels for shifts). In Section 3, some results related to a real dataset are also presented.

**Table 2.** Inner and outer reliabilities computed for the simulated Cases 1, 2, and 3. In the second column, the subset of TPs adopted for evaluating the inner/outer reliabilities are described using the following symbols: 'MS' is the subset of TPs shared between the 'master' and one or more 'slaves'; 'SS' is the subset of TPs shared between two or more 'slaves'; 'All' is the subset including all types of TPs.

| Case | TP Subset | $\sigma_{im}$ (pix) | Inner Reliability | | Outer Reliability | | |
|------|-----------|---------------------|-------------------|---------|-------------------|------------|-----------------------|
| | | | Max-Min Range (pix) | Average (pix) | Shifts (pix) | Scale (‰) | Rotation Angle ($10^{-3}$ gon) |
| 1 | MS | 1 | 4.2–4.4 | 4.3 | 0.27 | 3.8 | 9.8 |
| 2 | MS | 1 | 4.1–4.2 | 4.2 | 0.20 | 2.6 | 7.0 |
| | SS | 1 | 5.0–5.2 | 5.1 | 0.14 | 1.8 | 5.6 |
| | All | 1 | 4.1–5.2 | 4.5 | - | - | - |
| 3 | MS | 1 | 4.1–4.2 | 4.2 | 0.15 | 2.2 | 6.4 |
| | SS3 | 1 | 5.0–5.1 | 5.1 | 0.08 | 1.3 | 3.8 |
| | SS6 | 1 | 4.5–4.8 | 4.6 | 0.07 | 0.8 | 2.3 |
| | All | 1 | 4.1–5.1 | 4.5 | - | - | - |

## 3. Application to Landsat-5/TM Imagery

### 3.1. Dataset and Data Processing

The MIRA algorithm has been tested on a Landsat-5/TM time series (Level 1 products) made up of eleven frames imaged over the city of Multan, Pakistan. The images were collected from February 1998 to December 1998 (ID1: 9 February; ID2: 23 March; ID3: 14 April; ID4: 16 May; ID5: 4 August; ID6: 20 August; ID7: 7 October; ID8: 23 October; ID9: 8 November; ID10: 24 November; and ID11: 10 December). The GSD (ground sample distance) is 30 m for all the available spectral bands except the thermal infrared channel (TIR—Band TM6), whose GSD is 120 m. Due to such a low geometric resolution, images from Band TM6 have not been considered during next processing stages.

Since the radiometric content may largely differ in different wavelengths of multispectral imagery (see Table 3), uneven results are expected during pairwise FBM aimed at extracting corresponding TPs. Indeed, different land-cover types and surface materials are characterized by a non homogeneous spectral response. Thus, they may exhibit different image contrast according to the specific wavelength. In addition, shorter wavelengths in the visible domain (mainly bands TM1/TM2) are more affected by atmospheric scattering. In a previous paper [44], the robustness of MIRA against atmospheric effects was demonstrated. For this reason, FBM has been applied to the images in all bands without any preprocessing to correct these effects. Table 3 shows the results in terms of image pairs that have been successfully matched in different bands, and the total number of extracted TPs. A single image pair is considered as 'successfully matched' if at least $N_{min}$ = 12 TPs have been found. As can be seen, all bands except TM4 show results that are very close to one another regarding the fraction of successfully matched image pairs, ranging between 96% and 100% of 55 total possible combinations. This achievement is also motivated by the large spatial overlap between all the images. In the case of Band TM4 (NIR—near infrared), changes in vegetation cover during different seasons of the year may be strongly correlated to the variation of NIR content of the images, resulting in problems when seeking for corresponding features.

After the pairwise FBM stage is accomplished, image pairs need to be linked together to find multiple TPs and to detect which image should be preferably used as 'master' for the time series. This selection is done by analyzing the pairwise connections between images that are summarized in the *connectivity matrices* for different bands displayed in Figure 3. According to the results obtained from the most wavelengths, Image ID8 has been selected as 'master' using the methodology reported in Section 2.4. The choice is motivated by the highest number of images that are connected to it, i.e., images that share at least $N_{min}$ = 12 TPs. Image ID8 is shown in Figure 5, where extracted TPs are also overlaid. In some bands, only two images are not directly connected to Image ID8. The poor results obtained with FBM in the case of Band TM4 have also reflected in the connectivity graph, which is split into two clusters (Figure 6). The lack of connections suggests that the time series cannot be processed as a whole when using TPs extracted in spectral Band TM4. Looking at the other wavelengths, the total number of extracted TPs after reordering spans from 47,812 (TM1) to 75,073 (TM3) in terms of image observations (Table 3). This variability shows that the number of extracted multiple features significantly depends on the adopted wavelength, even though TPs are always sufficient to compute the image registration parameters. In Figure 7 three patches of the images from the 'Multan' dataset are shown before and after the alignment obtained using MIRA procedure.

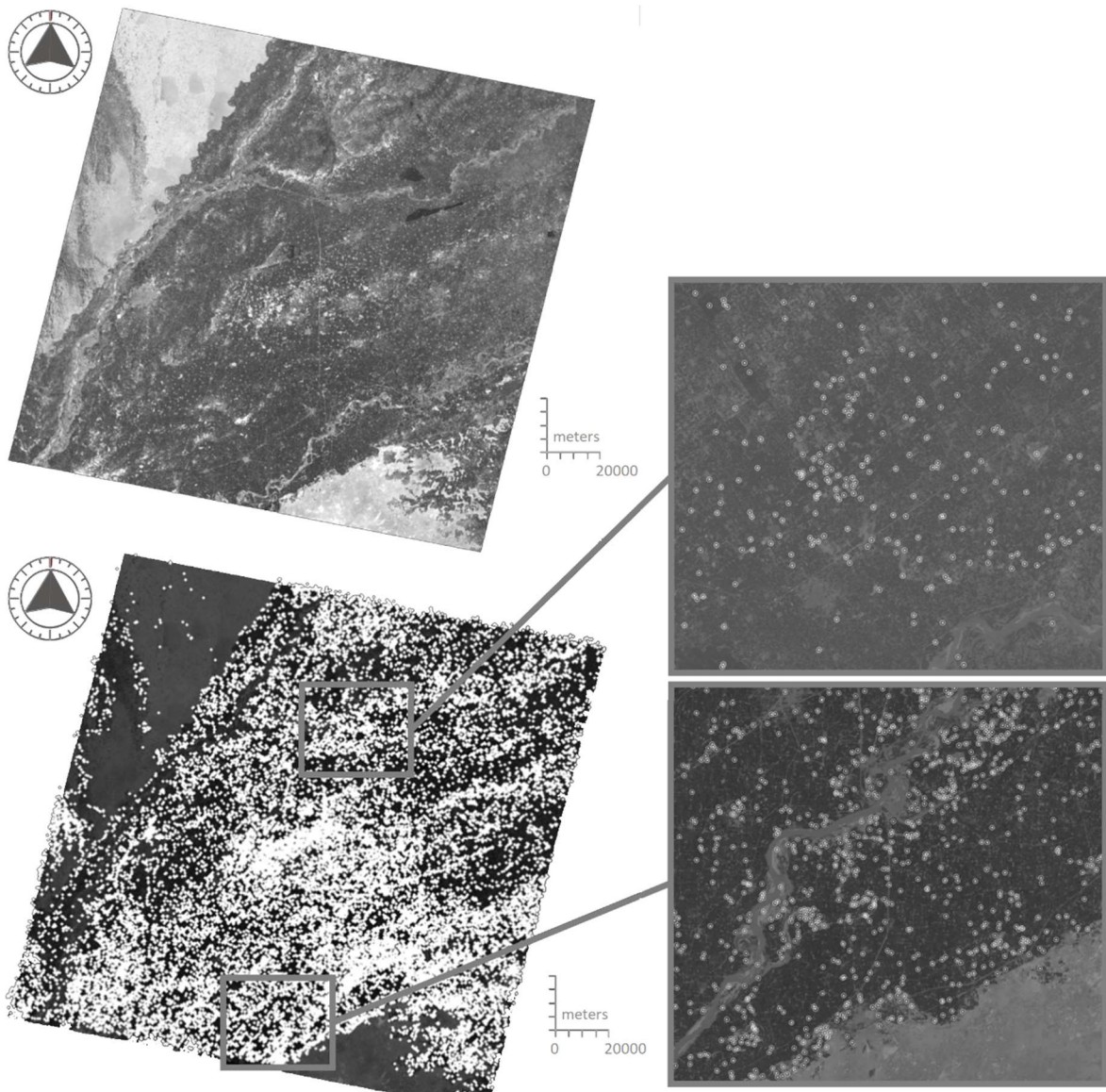

**Figure 5.** In the upper-left subfigure, the master image (ID8) of 'Multan' dataset is shown. In the lower-left subfigure, the same image is overlaid with all the TPs extracted with MIRA procedure. Also, those features that have not been found in the 'master' but they connect other images of the same dataset have been re-projected over the master. The two zoom-in windows on the right side display some details of the extracted features in two specific areas.

In Table 4 the focus is given to the multiplicity of extracted TPs, which directly reflects in the inner reliability of the observations. For simplicity, here the results obtained with spectral Band TM3 have been reported, since this band has provided the largest number of TPs. Similar outcomes have been also found from other spectral bands (except TM4). While the majority of TPs (approx. 70% of the total) are measured on two images only, the remaining 30% comprehend manifold points, i.e., TPs observed on at least three images. As expected, the most numerous classes are the ones including TPs visible on three (approx. 18%) and four images (approx. 7%), while the number of TPs dramatically drops down when considering higher multiplicities. It should be observed that a small group of TPs are visible on more than seven images.

**Table 3.** Summary of FBM results with images in different spectral bands of 'Multan' dataset. The number of image pairs that have been successfully matched refers to those image pairs sharing a sufficient number ($N_{min}$ = 12) of corresponding features after FBM. The percentage represents the number of image pairs that have been successfully matched over the total number of potential image combinations (n.a.: not applicable).

| Band | Spectral Content (nm) | Image Pairs Successfully Matched | | Total Number of Extracted TPs (after Reordering) |
| --- | --- | --- | --- | --- |
| | | (#) | (%) | # Matches |
| TM1 | Blue (0.45–0.52) | 53 | 96 | 47,812 |
| TM2 | Green (0.52–0.60) | 55 | 100 | 64,984 |
| TM3 | Red (0.63–0.69) | 54 | 98 | 75,073 |
| TM4 | NIR (0.76–0.90) | 14 | 25 | 16,294 |
| TM5 | SWIR (1.55–1.75) | 54 | 98 | 55,893 |
| TM6 | TIR (1.75–2.08) | n.a. | n.a. | n.a. |
| TM7 | SWIR (2.08–2.35) | 53 | 96 | 75,057 |

**Table 4.** Summary of TPs extracted in 'Multan' dataset with their multiplicity for both automatic data processing (MIRA) and manual measurement. Here results obtain in spectral Band TM3 are reported. TPs are considered with their multiplicity as re-projected on the space of the 'master' image.

| # Images | Number of Matched TPs in Band TM3 | | | |
| --- | --- | --- | --- | --- |
| | MIRA (Automatic) | | Manual | |
| | # Points | % Total | # Points | % Total |
| 2 | 26,138 | 69.63 | 1 | 0.95 |
| 3 | 4579 | 18.30 | 4 | 5.71 |
| 4 | 1300 | 6.93 | 5 | 9.52 |
| 5 | 486 | 3.24 | 3 | 7.14 |
| 6 | 148 | 1.18 | 4 | 11.43 |
| 7 | 53 | 0.49 | 2 | 6.67 |
| 8 | 19 | 0.20 | 6 | 22.86 |
| 9 | 1 | 0.01 | 5 | 21.43 |
| 10 | 1 | 0.01 | 3 | 14.29 |
| Total | 37,275 | 100.00 | 33 | 100.00 |

Table 5 shows some figures illustrating the ratio between equations and unknowns of the system adopted to estimate the image registration parameters and the coordinates of all TPs re-projected on the 'master' image space. In any spectral bands, this ratio is over three, resulting in a good global redundancy. In the case of Band TM4 as well, whose processing has been more problematic, the ratio equations/unknowns has been found close to three (2.8). As can be seen, subpixel accuracy has been obtained for all the spectral bands, as pinpointed by the values of estimated $\sigma_0$, which can be assumed as metrics for the average theoretical accuracy of measured image coordinates. Results achieved with visible bands (TM1, TM2, TM3) have been slightly better ($\sigma_0 \cong 0.5$ pixels) than the ones from short-wave infrared (TM5, TM7), which yielded $\sigma_0 \cong 0.6/0.7$ pixels. As mentioned before, NIR (TM4) has output two different clusters of images. Thus, the system as a whole has shown a rank deficiency and could not be solved together.

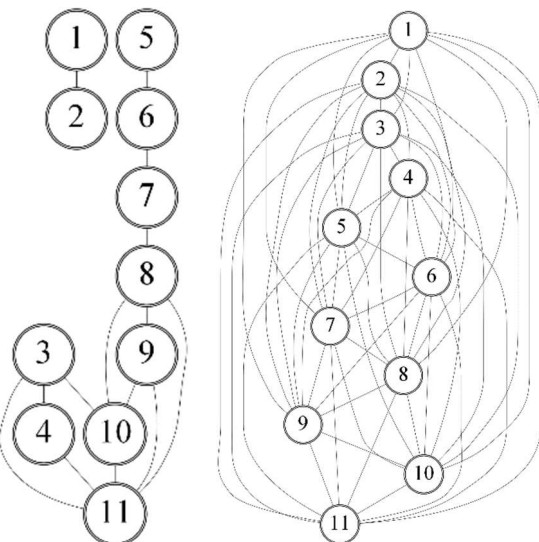

**Figure 6.** Connectivity graph obtained from Band TM4 (on the left) of the 'Multan' dataset, showing the separation of the entire blocks in two parts. This graph is compared to the one obtained from band TM3 (on the right), which corresponds to the best connections. In these graphs, circles represent the images while lines indicate the availability of sufficient TPs to connect each pair.

**Table 5.** Summary of global registration results after BBA in different spectral bands (n.a.: not applicable; r.d.: rank deficiency).

| Band | Spectral Content (nm) | Mode | Observation Equations (#) | Unknowns (#) | Global Redundancy (#) | Ratio Obs. eq.s/unk.s | $\sigma_0$ (pix) |
|------|------------------------|------|---------------------------|--------------|------------------------|------------------------|------------------|
| TM1 | Blue (0.45–0.52) | MIRA | 95,624 | 29,814 | 65,810 | 3.2 | 0.53 |
| TM2 | Green (0.52–0.60) | MIRA | 129,968 | 35,810 | 94,158 | 3.6 | 0.52 |
| TM3 | Red (0.63–0.69) | MIRA | 150,146 | 42,858 | 107,288 | 3.5 | 0.52 |
| | | Manual | 420 | 70 | 350 | 6.0 | 1.15 |
| TM4 | NIR (0.76–0.90) | MIRA | 32,588 | 11,712 | 20,876 | 2.8 | r.d. |
| TM5 | SWIR (1.55–1.75) | MIRA | 111,786 | 35,198 | 76,588 | 3.2 | 0.69 |
| TM6 | TIR (1.75–2.08) | MIRA | n.a. | n.a. | n.a. | n.a. | n.a. |
| TM7 | SWIR (2.08-2.35) | MIRA | 150,114 | 42,364 | 107,750 | 3.5 | 0.63 |

A set of estimated parameters $(a_j, b_j, c_j, d_j)$ for the 2D similarity transformation mapping each image j to the reference datum has been obtained per each spectral band. Since all bands are already mutually co-registered, the use of a specific band should provide the same parameters as in the case of others. The computed unknown parameters for all the spectral bands are graphically displayed in Figure 8, where it can be seen that similar results have been obtained. On one side, parameters $a_j$ and $b_j$ that provide information about scale and rotation, have been estimated for all the images as $a_j \cong 1$ and $b_j \cong 0$. These values demonstrate that all the images have neither scale variation nor rotation. On the other side, results for shift parameters $(c_j, d_j)$ have been different for individual images, as obvious, but consistent values have been obtained from different spectral bands. Indeed, the root mean square (RMS) of their variations have resulted approximately 0.05 pixels in both directions. This result shows that for the 'Multan' dataset all Landsat-5/TM spectral bands may be potentially used for image alignment, exception made for Bands TM4 and TM6.

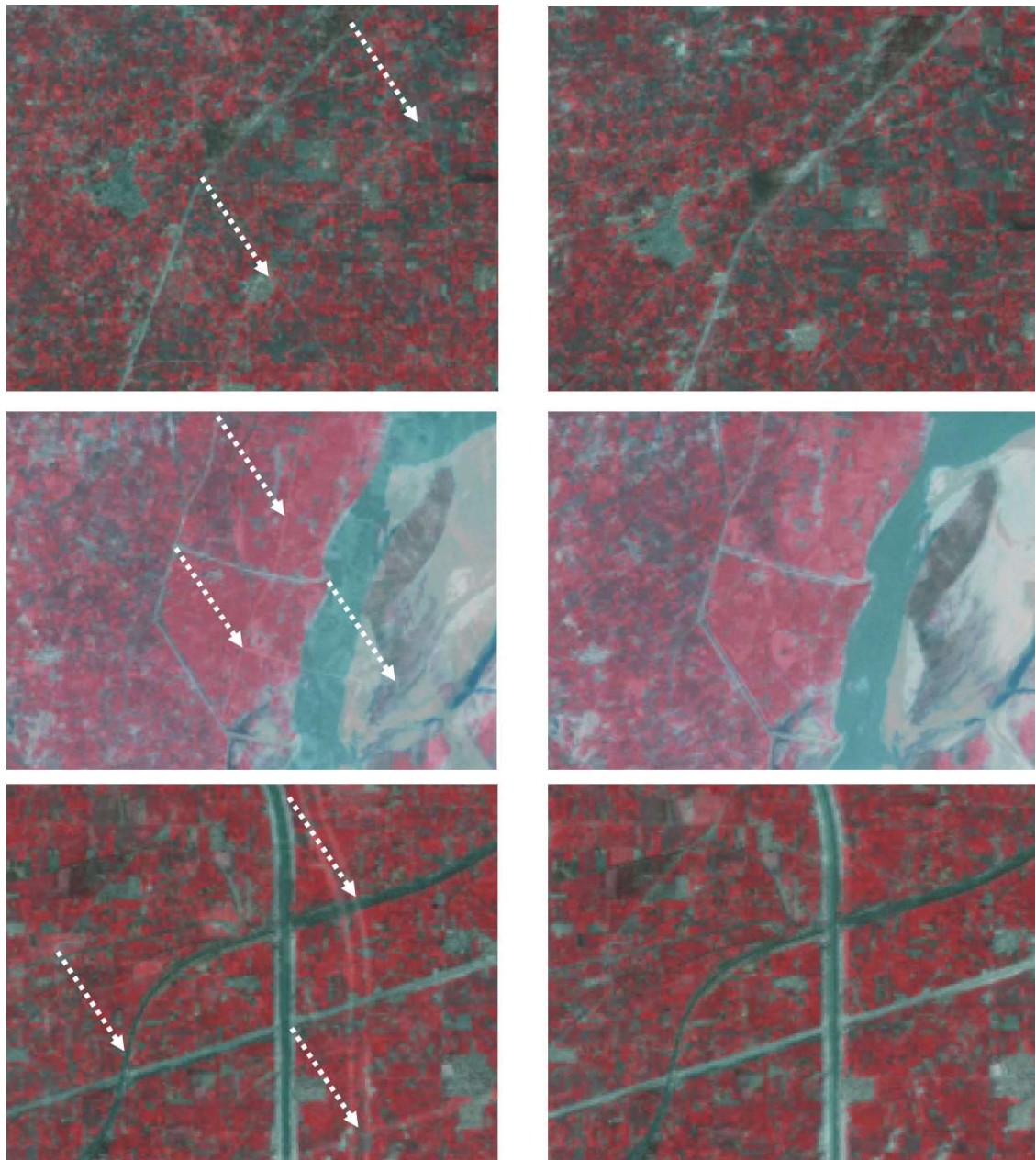

**Figure 7.** Overlap between unregistered (on the left column) and registered (on the right column) images of the 'Multan' dataset as obtained by applying MIRA procedure. Yellow arrows refer to the position of some corresponding features.

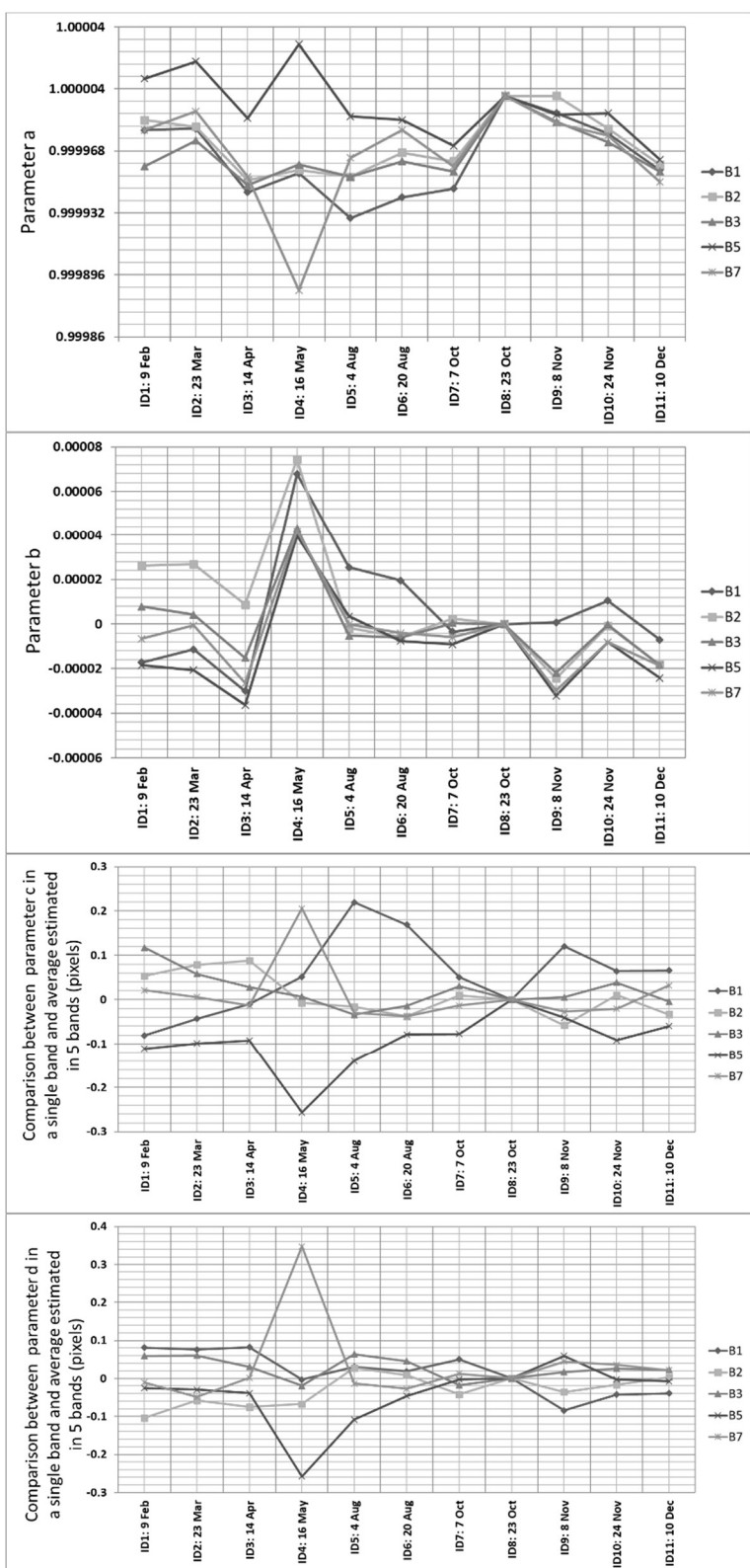

**Figure 8.** Comparison between the estimated parameters for all the images of 'Multan' dataset as obtained from different spectral bands (see legends on the right side of each graph). For parameters a and b the values obtained from three different experiments are displayed, while for parameters c and d (shifts) the differences with respect to the average estimated values are shown.

*3.2. Validation*

A first evaluation of the results described in the previous section has been based on the estimated theoretical accuracy and the consistency of the outcomes achieved in different spectral bands (see Section 3.1). On the other hand, a further assessment has been accomplished by comparing these results with the ones obtained from different methods applied to the same dataset:

1.　A standard 'slave-to-master' registration approach based on automatic measurements; and
2.　A multi-image approach based on BBA but using TPs that have been manually measured by a human operator.

3.2.1. Comparison between MIRA and 'Slave-To-Master' Registration

The goal of this experiment has been to compare the traditional 'slave-to-master' approach for image registration and MIRA. To this purpose, all the images have been directly aligned to the same 'master' (ID8) selected within MIRA application. The pairwise corresponding points obtained from FBM have been considered for computing the registration parameters of 2D similarity transformations between each 'slave' and the 'master'. In this case, the registration of each individual image is completely independent from the others. Some statistics on the results are shown in Table 6.

As it can be seen, the estimates of shift parameters have shown an average variation in the order of 0.15 pixels in x direction and 0.08 pixels in y direction, with maximum absolute differences of 0.39 pixels and 0.17 pixels in the case of Images ID1 and ID2, respectively. No relevant differences of scale and rotation have been found between both methods. It is interesting to notice that the largest variations have been found in the case of Images ID1 and ID2. This outcome might be motivated by the longer elapsed time between the 'master' (recorded on October) and these two images (recorded on February and March of the same year, respectively), which have resulted in large changes in the image content due to the different seasons.

**Table 6.** Comparison between results obtained with the 'slave-to-master' and the MIRA approach for image registration on 'Multan' dataset (ppm: part-per-million). In both cases automatically-extracted TPs have been used.

| Parameters | RMS of Variations on All the Images | Max Absolute Variations | Images Corresponding to Max Variations |
|---|---|---|---|
| Scale | 16 ppm | 33 ppm | ID2 |
| Rotation angle | $2.3 \times 10^{-3}$ gon | $1.2 \times 10^{-3}$ gon | ID2 |
| Shift x | 0.15 pixels | 0.39 pixels | ID1 |
| Shift y | 0.08 pixels | 0.17 pixels | ID2 |

3.2.2. Comparison between MIRA and Manual Registration

The main difference between automated (MIRA) and manual measurements is the number of extracted corresponding features. The automatic MIRA method has found 75,073 corresponding features in the images after pairwise FBM (see Table 3), corresponding to 37,275 different TPs after re-ordering. Manual measurements have provided only a few tens TPs, as shown in Table 4. Here, the number is limited by the operator's skill and time. This means that the ratio equations/unknowns and therefore the size of the design and normal matrix are very different in both cases. Table 5 shows the 'manual' results obtained with band TM3. This band has been chosen for manual processing because of the highest number of extracted features when using MIRA method.

Table 4 shows a comparison between the 'visibility' of the corresponding features in two or more images of the time series for both MIRA and manual measurements. The human operator only identified 210 corresponding features (in one working day), corresponding to only 33 multiple features, since they have been measured in several images.

Another critical difference is the size of the normal matrix **N**. In the case of manual measurements, a higher fraction of manifold TPs connecting the 'master' image with multiple 'slaves' has been measured. The number of 'slave-to-slave' TPs is lower. This gives to the **N** matrix a more compact form around the main diagonal, which may help reduce the computational cost. With the automatic MIRA method, the matrix **N** is instead very sparse as there are many more TPs also connecting 'slaves' images among them. This means that more off-diagonal non-zero elements are present.

Table 7 summarizes the average accuracy of the MIRA method compared to manual measurements. The departure between the average differences of shift vectors is 0.12 pixel in both x and y directions, whereas it is evident there is neither rotation nor scale variation for all the images of the dataset. The theoretical standard deviations achieved with the automated measurements is better (RMS = 0.06 pixels for shifts) than the one obtained with the manual method (RMS = 0.12 pixels for shifts), notwithstanding results are at subpixel level in both cases. The larger number of observations obtained with the MIRA method may be addressed as the main reason for the better theoretical accuracy.

**Table 7.** Comparison between results obtained with manual and automatic (MIRA) processing of 'Multan' dataset.

| Parameters | Average Discrepancies between Estimated Parameters (Manual vs. MIRA) | RMS of Estimated Theoretical Accuracy | |
|---|---|---|---|
| | | Manual | MIRA |
| Scale | $2.0 \times 10^{-5} \pm 2.7 \times 10^{-5}$ | $6.1 \times 10^{-5}$ | $2.1 \times 10^{-5}$ |
| Rotation angle | $1.1 \times 10^{-5} \pm 1.2 \times 10^{-3}$ gon | $3.8 \times 10^{-3}$ gon | $1.3 \times 10^{-3}$ gon |
| Shift x | $0.02 \pm 0.15$ pixels | 0.12 | 0.06 |
| Shift y | $0.07 \pm 0.09$ pixels | 0.12 | 0.06 |

## 4. Discussion

The quality of image registration obtained from the MIRA method may be evaluated by considering three different aspects: the estimated value of the variance of unit weight observations ($\hat{\sigma}_0^2$), the comparison against results obtained from 'slave-to-master' registration based on automatically extracted TPs, and the results obtained using manual measurements of TPs, but within a BBA solution.

The estimated $\hat{\sigma}_0^2$ after BBA based on observation extracted with MIRA has resulted in the order of approximately half pixel size, with small variations depending on the adopted spectral band (see Table 5). The achieved subpixel value has confirmed the good fit with the adopted geometric model implemented for image registration (2D similarity) as well as the precision of automatically extracted TPs remained after data snooping. In Figures 9 and 10 the results achieved with all three applied registration methods are graphically compared. No significant variations may be noticed about rotations and scales, which can be derived from parameters $a_j$ and $b_j$. This is due to the imaging scheme of Landsat: fixed nadir-looking. Thus, images of the same site, but collected at different times, will have tiny variations in image scale and very small rotations. However, when processing time series imaged with different viewing angles, image rotations and scale variations could not be neglected. That is the typical case of high-resolution satellite images, but also that of multi-sensor datasets, including the data processing of radar images, which are usually collected with different incidence viewing angles and/or different orbits [45,46]. On the other hand, subpixel discrepancies have been found for shifts (parameters $c_j$ and $d_j$,) in the Landsat time-series. The consistency between all these outputs confirms the correctness of MIRA approach.

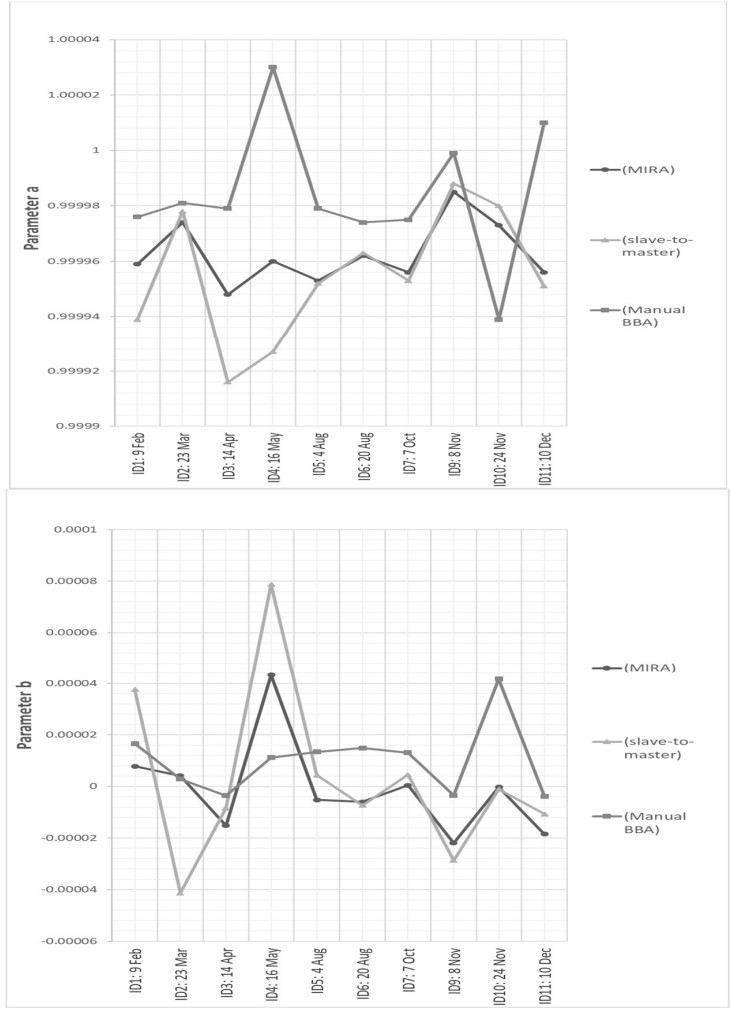

**Figure 9.** Comparison between the estimated parameters for all the images of 'Multan' dataset (band TM3) obtained by using MIRA and other methods: 'slave-to-master' (see Section 3.2.1) and 'Manual BBA' (see Section 3.2.2). For parameters a and b the values obtained from three different experiments are shown.

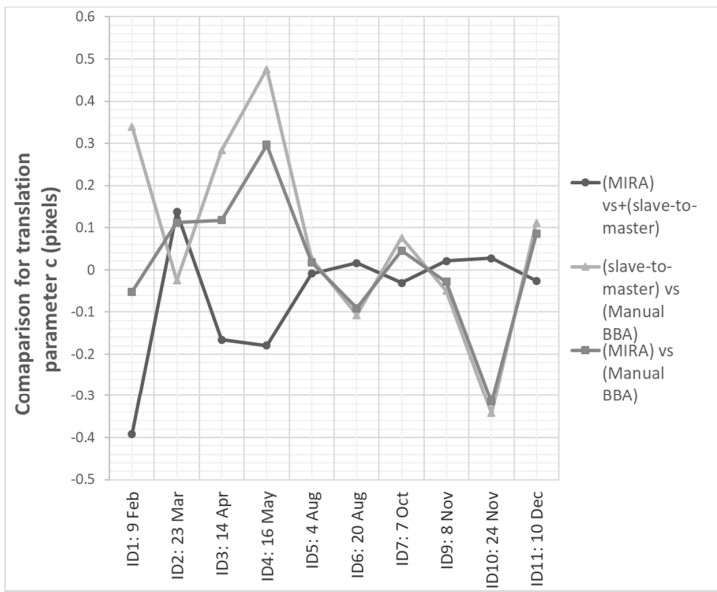

**Figure 10.** *Cont.*

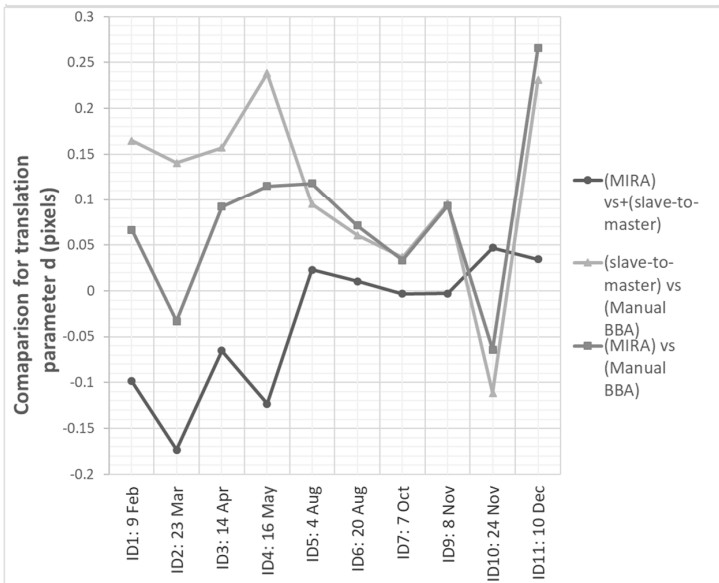

**Figure 10.** Comparison between the estimated parameters for all the images of 'Multan' dataset (band TM3) obtained by using MIRA and other methods: 'slave-to-master' (see Section 3.2.1) and 'Manual BBA' (see Section 3.2.2). For parameters c and d (shifts) the pairwise differences are shown.

Inner and outer reliabilities have been computed for all three processing methods based on MIRA and manual observations. Values are reported in Table 8, where the results based on automatically extracted TPs have been also recomputed using the same precision of 'manual' measurements, which is supposed to be ±1 pixel. This solution has allowed to normalize the results and to make them comparable, since the TP precision had a linear scaling effect. In fact, the theoretical accuracy of MIRA observations has been estimated in the order of 0.5 pixels, which is consistent with the expected precision of FBM [33]. The analysis of *inner reliability* clearly highlights that the minimum size of detectable errors during data snooping significantly decreases when moving from 'slave-to-master' to BBA approach. In the former case, the inner reliability of twofold TPs (i.e., the ones visible on two images) is in the order of 6.1 pixels, while it improves up to 4.1 pixels when considering multiple TPs that can be obtained from BBA. As can be seen in Table 8, the inner reliabilities do not depend only on the multiplicity and the precision of the considered TP, but also on the fact that this is shared with the 'master' image or not (see the second column). These discrepancies are in the order of a few tenths pixels and are motivated by the fact that the observation equations implemented in BBA are always referred to the 'master'. When looking at the *outer reliability*, a first comment should concern the evident advantage of using the large number of TPs extracted by the automatic MIRA procedure. Indeed, the negative effect of a maximum undetected error, which is indicated by the inner reliability value, is mitigated by the redundancy of the observations. Consequently, similar values for the inner reliability for manual and automatic measurements (in the range between 4–6 pixels), when they are processed within the BBA, may result in errors on the final shift estimates up to approximately 1.5 pixels in the former case and 0.04 pixels in in the latter case, respectively. The differences of scaling and rotation parameters are less influenced.

**Table 8.** Inner and outer reliabilities computed for the 'Multan' dataset. In the second column, the subset of TPs adopted for evaluating the inner/outer reliabilities are described using the following symbols: 'MS' is the subset of TPs shared between the 'master' and one or more 'slaves'; 'SS' is the subset of TPs shared between two or more 'slaves'; 'All' is the subset including all types of TPs.

| Case | TP Subset | $\sigma_{im}$ (pix) | Inner Reliability | | Outer Reliability | | |
|---|---|---|---|---|---|---|---|
| | | | Max-Min Range (pix) | Average (pix) | Shifts (pix) | Scale (‰) | Rotation ($10^{-3}$ gon) |
| 'Slave-to-master' | MS | 0.5 | 3.0–3.1 | 3.0 | 0.033 | 0.006 | 0.31 |
| MIRA | MS | 0.5 | 2.0–2.1 | 2.0 | 0.022 | 0.004 | 0.21 |
| | SS | 0.5 | 2.1–3.1 | 2.8 | 0.020 | 0.004 | 0.23 |
| | All | 0.5 | 2.0–3.1 | 2.4 | - | - | - |
| 'Slave-to-master' | MS | 1 | 6.1–6.2 | 6.1 | 0.067 | 0.011 | 0.63 |
| Manual BBA | MS | 1 | 4.1–4.8 | 4.2 | 1.56 | 0.26 | 15.0 |
| | SS | 1 | 4.5–6.0 | 4.9 | 1.49 | 0.25 | 14.6 |
| | All | 1 | 4.1–6.0 | 4.4 | - | - | - |
| MIRA | MS | 1 | 4.0–4.1 | 4.0 | 0.044 | 0.008 | 0.42 |
| | SS | 1 | 4.2–6.1 | 5.6 | 0.040 | 0.008 | 0.46 |
| | All | 1 | 4.0–6.1 | 4.8 | - | - | - |

These results conclude that the high data redundancy and the improved inner/outer reliability that may be obtained when using MIRA are two fundamental properties supporting the use of such an automatic procedure. The high redundancy allows to mitigate the degrading effect of residual measurement errors. The low values for the inner reliability may support the chance to limit the size of undetected errors. Of course, this second advantage depends on the fact that a small number of residual outliers are input in the BBA observation dataset. This chance is supported by the preliminary application of multiple scrutinizing technique to detect outliers during the FBM stage, which is supposed to leave a small number of outliers.

It should be also recalled that the other important advantage of the MIRA procedure is given by the chance to register possible images that are not directly connected to the 'master' because they do not individually share enough TPs with it. If these images may be linked to other images in the block that are connected to the 'master', the BBA solution allows to compute the registration of the whole dataset. This alternative solution is not possible using traditional 'slave-to-master' registration approach.

## 5. Conclusions

This paper presented some important developments of an automatic method (multi-image robust alignment—MIRA) for the registration of remotely sensed time series, where multiple images are simultaneously registered in a bundle block adjustment fashion after the extraction of corresponding features using feature-based matching (FBM).

This approach has two main advantages if compared to standard 'slave-to-master' registration methods. The first consists in the chance to align also those images without direct connection with the 'master', which in the MIRA procedure is only adopted for setting up the spatial datum. The second is the higher reliability of the solution since the redundancy of the observations is fully exploited. While in a previous paper [27] the basic concept of this approach was presented, here the focus is given to the automation of the whole procedure, which requires a high-degree of robustness against blunders, the availability of objective parameters and criteria to support decisions within the process, a rigorous stochastic formulation for the observation equations in least-squares adjustment, and the presence of intermediate quality checks (e.g., after pairwise FBM).

A set of 2D polynomial transformations is available to better fit different datasets and images from diverse sensors. Consequently, MIRA may be successfully applied to medium-resolution satellite data (GSD between 10 m and 30 m), while its implementation with high- and very high-resolution

imagery still needs additional development to integrate more suitable geometric models, e.g., rational polynomial functions or physical sensor models [47]. Anyway, it should be mentioned that with more involved geometric models that require a larger number of parameters, the reliability analysis that has been proposed here may be less meaningful. On the other hand, 2D polynomial transformations are sufficient for the registration of medium-resolution satellite images, such as the ones derived from NASA Landsat, ESA Sentinel-2 platform, and the British Disaster Monitoring Constellation. These datasets and other similar ones, which may be expected in the future, are highly prone to be exploited for hyper-temporal remote sensing with very short revisit time (a few days between observations). The MIRA procedure may play a vital role in the automatic subpixel alignment of such datasets.

In the perspective of processing huge datasets as well, the large size may create some problems in the inversion of the normal matrix **N**, due to the high computational cost of this operation. This would prevent the computation of the covariance matrix of the solution, and then the evaluation of the theoretical accuracy of estimated parameters as well as the redundancy matrix that is necessary for the reliability analysis [48]. To overcome this shortcoming, an ad hoc procedure for the decimation of the corresponding features based on their image multiplicity will be implemented in future developments.

At the moment, the selection of the spectral band to be used for the image alignment is left to the user. In the considered case study, similar results regarding the achievable precision have been obtained from different wavelengths, with the exception of near and thermal infrared. On the other hand, combining the point correspondences obtained in different bands may also be an opportunity to extend the use of the MIRA procedure, especially when different types of images should be registered together.

**Author Contributions:** Conceptualization: M.S.; data curation: L.B.; funding acquisition: M.G.; investigation: L.B.; methodology: L.B.; software: L.B.; supervision: M.S. and M.G.; validation: L.B.; writing—original draft: L.B.; writing—review and editing: M.S. and M.G.

**Funding:** This research was funded by the Italian Ministry of Education, University, and Research (MIUR) within the grant FIRB 2010 entitled: "Subpixel techniques for matching, image registration and change detection with applications to civil and environmental engineering".

**Acknowledgments:** The authors would like to acknowledge NASA for the availability of Landsat images used in the experiments.

**Conflicts of Interest:** The authors declare no conflict of interest.

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
