# Peer review of "Multi-Image Robust Alignment of Medium-Resolution Satellite Imagery"

_remotesensing, doi:10.3390/rs10121969_

Round 1
Reviewer 1 Report
In this paper, the author proposed an automatic Multi-Image Robust Alignment (MIRA) procedure to simultaneously co-register a time series of medium-resolution satellite images in a bundle adjustment process.
Major Concerns:
1. The title is quite confusing. Initially, when reading the title, I thought the bundle block adjustment is used for 3D reconstruction from multiple satellite images. Then, I noticed that the proposed BBA is only used for the registration of a time series of satellite images. It is better to directly highlight the objective of image registration in the title.
2. I’m not sure if the author is familiar with the Structure-from-Motion (SfM) process, which simultaneously automates the process of image orientation recovery and 3D structure reconstruction. Similar to the proposed MIRA approach, SfM-based approaches are usually conducted in three steps. In the first step, stereo-based relative orientation parameters are estimated through the identified feature correspondences among overlapping images. Then, the exterior orientation parameters of all the involved imagery are recovered in the same reference coordinate system. Finally, a bundle adjustment is conducted to refine all parameters. The proposed slave-to-master and slave-to-slave image associations are quite similar to the strategy used in the existing SfM approaches.
3. Image registration/stitching while using automated feature matching and bundle adjustment is not a new topic. I’m not sure the utilization of “slave-to-slave” matches in the registration of multiple satellite images is a sufficient contribution for a journal publication.
Other comments:
1. Line 137: Why n+1 elements in the data set instead of n images? If the dataset is made of one reference image and n slave images, this has to be explicitly explained before introducing the total number of elements in the dataset.
2. Figure 2 “Creation of n+1 Image Pair Combinations”: n+1 or n(n+1)/2 combinations?
3. Line 158: The number of SIFT and SURF features depends on the utilized thresholds (e.g., the threshold for selecting peaks in the generated Difference of Gaussians). I’m not sure if it is appropriate to claim that SURF returns fewer points when compared to SIFT.
4. Line 169: smallest distance or largest distance? The candidate pair of matches should have the smallest Euclidean distance between descriptors.
5. Line 170: ratio test: first distance / second distance?
6. Lines 178 to 180: Please provide mathematical formulas for the utilized mapping model (i.e., similarity, 1st degree, 2nd degree, and 3d degree polynomials). Instead of using the proposed model, why not use an affine transformation for the registration of images.
7. Equation 4: What is s(n+1)? I don’t understand how Equation 4 guarantees the selected ‘master’ image can maximize the number of images that share enough tie points with it.
8. Provide figures to show the time series of satellite images after registration?
Author Response
First of all we would like to thank Reviewer 1 for the useful comments he/she provided. In this paper, the author proposed an automatic Multi-Image Robust Alignment (MIRA) procedure to simultaneously co-register a time series of medium-resolution satellite images in a bundle adjustment process. Major Concerns: 1. The title is quite confusing. Initially, when reading the title, I thought the bundle block adjustment is used for 3D reconstruction from multiple satellite images. Then, I noticed that the proposed BBA is only used for the registration of a time series of satellite images. It is better to directly highlight the objective of image registration in the title. Answer: We agree with this comment and changed the title accordingly into: “Multi-Image Robust Alignment of Medium-Resolution Satellite Imagery.” In addition, this new title also incorporates the name of the procedure we developed (MIRA). 2. I’m not sure if the author is familiar with the Structure-from-Motion (SfM) process, which simultaneously automates the process of image orientation recovery and 3D structure reconstruction. Similar to the proposed MIRA approach, SfM-based approaches are usually conducted in three steps. In the first step, stereo-based relative orientation parameters are estimated through the identified feature correspondences among overlapping images. Then, the exterior orientation parameters of all the involved imagery are recovered in the same reference coordinate system. Finally, a bundle adjustment is conducted to refine all parameters. The proposed slave-to-master and slave-to-slave image associations are quite similar to the strategy used in the existing SfM approaches. Answer: Honestly, the authors are quite familiar with the SfM, as witnessed by some papers they have published. Among all, Barazzetti L., Scaioni M., Remondino, F., 2010. “Orientation and 3D modelling from markerless terrestrial images: combining accuracy with automation.” The Photogrammetric Record, Vol. 25(132), pp. 356-381, where we tried to bring the SfM into the photogrammetric community, and then into a wider application domain. For this reason, the submitted version of the paper already reported the following comments that is supportive of Reviewer 1's opinion (see Subsect. 2.4): “The adopted procedure is similar to the scheme followed in the ‘Structure-from-Motion’ technique [40] for the orientation of image blocks in close-range photogrammetry. By looking at the connectivity graph, all ‘slaves’ that are directly linked to the ‘master’ image may be approximately registered in pairwise, independent manner. In this way, any images in this first group of registered images can be used as new ‘master’ images to register other ‘slave’ images that would share sufficient corresponding points with one of them. By using such approximate registration parameters is then possible to re-project the coordinates of any points on the main ‘master’ image space.” 3. Image registration/stitching while using automated feature matching and bundle adjustment is not a new topic. I’m not sure the utilization of “slave-to-slave” matches in the registration of multiple satellite images is a sufficient contribution for a journal publication. Answer: We agree that in some applications (e.g. SfM with terrestrial or UAV images) the use of additional slave-to-slave matches is quite common. On the other hand, this is not the case of remotely sensed data, in which most applications are carried out with a basic master-to-slave processing. At the present, there is no remote sensing sw able to process such additional combinations. Other comments: 1. Line 137: Why n+1 elements in the data set instead of n images? If the dataset is made of one reference image and n slave images, this has to be explicitly explained before introducing the total number of elements in the dataset. Answer: In the first version of the manuscript we meant the block consisted of n “slaves” and 1 “master” image. Anyway, we agreed with Reviewer 1 that this solution is confusing and we have turned it down to n images in total. 2. Figure 2 “Creation of n+1 Image Pair Combinations”: n+1 or n(n+1)/2 combinations? Answer: We would like to thank Reviewer 1 for this correction. Now Figure 2 has been corrected by considering the new total number of images n instead of n+1. 3. Line 158: The number of SIFT and SURF features depends on the utilized thresholds (e.g., the threshold for selecting peaks in the generated Difference of Gaussians). I’m not sure if it is appropriate to claim that SURF returns fewer points when compared to SIFT. Answer: We agreed with this comment. Thus we removed this statement from the manuscript. 4. Line 169: smallest distance or largest distance? The candidate pair of matches should have the smallest Euclidean distance between descriptors. Answer: Corrected as suggested. 5. Line 170: ratio test: first distance / second distance? Answer: We may confirm that the Ratio Test works as Reviewer 1 mentioned. We included th suggested explanation in the line. 6. Lines 178 to 180: Please provide mathematical formulas for the utilized mapping model (i.e., similarity, 1st degree, 2nd degree, and 3d degree polynomials). Instead of using the proposed model, why not use an affine transformation for the registration of images. Answer: we included in Table 1 the mathematical formulas of those mapping models that can be applied. Thank you for pointing out the definition of mapping functions may lead to confusion. That could be caused by the use of different terms among different communities (e.g. Remote Sensing, Photogrammetry, Computer Vision). Besides, we know that in the Remote Sensing community users of Harris’ ENVI could be puzzled by the unusual implementation of the 1st degree polynomial transformation - called first order polynomial warping in ENVI - which is not a 1st degree polynomial but uses the following transformation (8 parameters): x = a1 + a2*X + a3*Y + a4*XY y = b1 + b2*X + b3*Y + b4*XY while the affine transformation - called RST warping in ENVI - is the standard 1st degree polynomial transformation (6 parameters): x = a1 + a2*X + a3*Y y = b1 + b2*X + b3*Y To avoid this confusion, we amended Table 1 by rephrasing “1st degree Polynomial (u=1)” into “1st degree Polynomial / Affine transformation (u=1)”. 7. Equation 4: What is s(n+1)? I don’t understand how Equation 4 guarantees the selected ‘master’ image can maximize the number of images that share enough tie points with it. Answer: Actually we have realized that the current explanation of the algorithm for selecting the “master” image is not clear. We tried to reformulate a better version by simply listing both criteria adopted to select the ‘master’ image. 8. Provide figures to show the time series of satellite images after registration? Answer: We understand the reviewer would like to see a figure with all the images of the time series after co-registration with MIRA. However, since in our case study we used Landsat-5/TM images (185 km width x 185 km height with 30 m spatial resolution), unfortunately these differences cannot be highlighted in the printed figures (approximately 1,000 pixels width/height at 300 dpi). On the other hand, we could provide figures showing the matched points in all the satellite images. However, these figures could be not informative, again due to the large image swath and small print size. That’s why we prefer showing results using the connectivity graph of Figure 6 and numeric tables.
Reviewer 2 Report
This paper presents algorithm called MIRA (Multi-Image Robust Alignment) for registering medium resolution satellite images. Experiments have been made using 3 simulated datasets and one real dataset. Results have been compared with 2 other methods, 'slave-to-master' and manual alignment, where proposed MIRA generally showed better accuracy.
There are some points that need to be explained:
1. In section 2.1, n+1 elements should have (n+1)*(n+2)/2 image pairs? Also, Figure 2 says there are n+1 image pair combinations?
2. Table 7 has 2 rows with the same name (manual)- probably one should be named MIRA?
3. When you have compared MIRA with other methods, actually in 2 cases (in real dataset, section 3) your method did not produce meaningful results. Would this also happen with other methods - e.g. that you have compared your method with? Can you explain?
4. Table 6 - cannot be seen what is being compared? It looks like only results for 'slave-to-master' registration is presented; maybe to show results for the proposed algorithm also?
5. Table 8 is referenced on page 18, but it appears on page 20. Can you move it near the place where it is firstly referenced?
6. You have shown in section 3 that proposed method has better accuracy for shifts in x and y direction. However, probably due to the used dataset, parameters a and b are more or less fixed (a=1, b=0). Can you show/conclude if your proposed method has better accuracy for scale and rotation angle parameters (a, b)?
Author Response
First of all, we would like to answer Reviewer 2 for his/her useful suggestions and comments.
This paper presents algorithm called MIRA (Multi-Image Robust Alignment) for registering medium resolution satellite images. Experiments have been made using 3 simulated datasets and one real dataset. Results have been compared with 2 other methods, 'slave-to-master' and manual alignment, where proposed MIRA generally showed better accuracy. There are some points that need to be explained: 1. In section 2.1, n+1 elements should have (n+1)*(n+2)/2 image pairs? Also, Figure 2 says there are n+1 image pair combinations? Answer: We modified the total number of images from n+1 to n. Thus the number of image pair combinations is n(n-1)/2. We have also corrected Figure 2 accordingly. 2. Table 7 has 2 rows with the same name (manual)- probably one should be named MIRA? Answer: actually we found a typos with the name of the rightmost column (‘manual’->’MIRA’) 3. When you have compared MIRA with other methods, actually in 2 cases (in real dataset, section 3) your method did not produce meaningful results. Would this also happen with other methods - e.g. that you have compared your method with? Can you explain? Answer: We would like to remark that our method does not always necessarily provide better results than others in term of registration precision. This achievement also depend on the quality of the images. If the quality is good, other methods also may provide satisfying results. The key-point of our method is in term of robustness/reliability, i.e., the higher chance to detect gross errors if present. 4. Table 6 - cannot be seen what is being compared? It looks like only results for 'slave-to-master' registration is presented; maybe to show results for the proposed algorithm also? Answer: Table 6 summarizes those results related to the test described in Par. 3.2.1. Here we wanted to compare the ‘slave-to-master’ registration approach and the multi-image adjustment implemented in MIRA. In both cases we considered tie points that were automatically extracted using the procedure described in this paper, but in the first we considered also direct connections to the ‘master,’ while in the second we considered all connections (i.e., also the slave-to-slave connections). 5. Table 8 is referenced on page 18, but it appears on page 20. Can you move it near the place where it is firstly referenced? Answer: Now we have moved Table 8 closer to the point where it is quoted. 6. You have shown in section 3 that proposed method has better accuracy for shifts in x and y direction. However, probably due to the used dataset, parameters a and b are more or less fixed (a=1, b=0). Can you show/conclude if your proposed method has better accuracy for scale and rotation angle parameters (a, b)? Answer: The reviewer is correct. The different accuracy of parameters is mainly due to the nature of the data used. Since Landsat images are collected with a fixed nadir-looking geometry, images of the same site but collected in different time will have tiny variations in image scale and very small rotations. When processing time series imaged with different viewing angles, results could be different. That is the typical case of high resolution satellite images, but also that of multi-angle sensors (such as the Multi-angle Imaging SpectroRadiometer MISR) or the case of multi-sensor data imaged with different satellites (e.g. Landsat-7/ETM+ and Landsat-8/OLI and Sentinel-2A and Sentinel 2-B). In the latter case, while all the four sensors are nadir looking, thus with small image scale variation and rotations, nevertheless Landsat and Sentinel have a different image scale (i.e. spatial resolution) and their images have a different footprint, thus could be rotated. Another scenario where the time series could include rotated images is when using images collected by satellites after their last spacecraft maneuver (a well-known case is that of Landsat-5/TM). Or when processing radar images which are usually imaged with different incidence viewing angles and/or different orbits (ascending vs descending). We have added two new references to our past works where we tested different imaging schemes (e.g. optical high-resolution tri-stereo images, radar high-resolution images with different incidence viewing angles, radar medium-resolution images with different orbits) and also rephrased the second paragraph of section Discussion as follows:“[...] No significant variations may be noticed about rotations and scales, which can be derived from parameters aj and bj. This is due to the imaging scheme of Landsat: fixed nadir-looking. Thus, images of the same site but collected in different time will have tiny variations in image scale and very small rotations. However, when processing time series imaged with different viewing angles, image rotations and scale variations could be not negligible. That is the typical case of high-resolution satellite images, but also that of multi-sensor data set, including the data processing of radar images which are usually collected with different incidence viewing angles and/or different orbits [45,46]. On the other hand, subpixel discrepancies have been found for shifts (parameters cj and dj,) in the Landsat time series. The consistency between all these outputs confirms the correctness of MIRA approach. [...]” [45] Gianinetto, M; Monno, V.; Barazzetti, L.; Dini, L.; Daraio, M.G.; Rota Nodari, F. Subpixel geocoding of COSMO-SkyMed and Sentinel-1 time series imaged with different geometry. Proceedings of 2016 International Geoscience and Remote Sensing Symposium, China, 10-15 July 2016, IEEE: Piscataway, USA, 2016, pp. 5007-5010, DOI: 10.1109/IGARSS.2016.7730306. [46] Gianinetto, M.; Barazzetti, L.; Dini, L.; Fusiello, A.; Toldo, R. Geometric registration of remotely sensed data with SAMIR. Proceedings of the 3rd International Conference on Remote Sensing and Geoinformation of Environment, Cyprus, 16-19 March 2015; Hadjimitsis, D.G.; Themistocleous, K.; Michaelides, S.; Papadavid, G., Eds.; SPIE: Bellingham, USA, 2015, Volume 9535, Article Number 95350Q, DOI: 10.1117/12.2192424.
Reviewer 3 Report
Paper presents an automatic Multi-Image Robust Alignment (MIRA) procedure to co-register a time series of medium-resolution satellite images in a bundle block adjustment (BBA) way.
Introduction is well presented. Nevertheless the methodology presenstation could be improved.
During 'validation (section 3.2.) you have used manual and automatic method to find the correspondences. I have a question if these tie points (features) are the same type of data/featrures/points?
If the image contrast has an influence on the 'matching' quality?
to discussion: what is the effective and parametric improvement of your feature. could you indicate the exact advantage of your method? If it is more precise (in what area) and how about the computational pros of this approach?
Please revise your paper, I have a feeling it is a little bit disordered. Espcecially in method description.
Other remarks:
Please consider unification of the font size. (example: lines: 162-165)
Please improve figure 8. Yhis font is hard to read and axis description is invisible.
Author Response
First of all, we would like to thank Reviewer 3 for his/her relevant comments and suggestions useful to improve the paper. Paper presents an automatic Multi-Image Robust Alignment (MIRA) procedure to co-register a time series of medium-resolution satellite images in a bundle block adjustment (BBA) way. Introduction is well presented. Nevertheless the methodology presentation could be improved. During 'validation (section 3.2.) you have used manual and automatic method to find the correspondences. I have a question if these tie points (features) are the same type of data/features/points? Answer: In section 3.2 we show the comparison between manual and automatic image registration. As described, the major difference is the number of corresponding extracted features (i.e., tie points): dozens for manual measures and dozens of thousand for automatic. That means the automatic tie points will not be the same tie points digitized by the operator for the manual check. Besides, we do not highlighted any relationship between tie points and land cover classes or landscape characterististics. Thus, in general all automatic image matching methods (not only MIRA) produce tie points which are different (for number, location and spectral properties) compared to the tie points that could be extracted by an expert operator. Of course, one could argue that manual measures might be more accurate than automatic measures. However, Table 7 summarizes the comparison between the MIRA method and manual measurements: the theoretical standard deviations achieved with MIRA are better than those of the manual measures because of the larger number of observations. In fact, if a human operator may better scrutinize the more suitable features to be measured as tie points, the automatic algorithms try to do that on the basis of some image properties, that may lead to extract other points. On the other hand, a basic rule of the automatica registration method is that the less individual accuracy of tie points is compensated by the large number of observations. If automatic measurements are not biased by systematic effects but are only affected by Normally-distributed random errors, the availability of a large number of observations helps improve the final precision of the registration process. If the image contrast has an influence on the 'matching' quality? Answer: The image contrast is one of the main properties influencing matching, but here the use of a robust Feature-based technique based on SURF may help mitigate the effect of image contrast changes. Anyway, in the Introduction we mention that: “Of course, in the case of poor image texture, the automatic extraction of TP’s may easily fail. This case frequently happens, for instance, when a significant portion of the image depicts a water body. On the other hand, this problem does not depend on the method used for the measurement of TP’s, since in the case of poor image texture also the interactive approach may result in severe problems.” to discussion: what is the effective and parametric improvement of your feature. could you indicate the exact advantage of your method? If it is more precise (in what area) and how about the computational pros of this approach? Answer: The advantages of our MIRA method are summarized in the end of the “Discussion” Section: “These results bring to conclude that the high data redundancy and the improved inner/outer reliability that may be obtained when using MIRA are two fundamental properties supporting the use of such an automatic procedure. The high redundancy allows to mitigate the degrading effect of residual measurement errors. The low values for the inner reliability may support the chance to limit the size of undetected errors. Of course, this second advantage depends on the fact that a small number of residual outliers are input in the BBA observation data set. This chance is supported by the preliminary application of multiple scrutinizing technique to detect outliers during the FBM stage, which is supposed to leave a small number of outliers.” We have also added the following lines to complete the description of the advantages; “It should be also recalled that the other important advantage of MIRA procedure is given by the chance to register possible images that are not directly connected to the ‘master’ images because they do not individually share enough TP’s with it. If these images may be linked to other images in the block that are connected to the ‘master’, the BBA solution allows to compute the registration of the whole data set. This alternative solution is not possible using traditional ‘slave-to-master’ registration approach.” Please revise your paper, I have a feeling it is a little bit disordered. Especially in method description. Answer: we have tried to better organize and explain some parts of the method description, as you may see from the yellow-highlighted lines. Other remarks: Please consider unification of the font size. (example: lines: 162-165) Answer: corrected as suggested. Please improve figure 8. Yhis font is hard to read and axis description is invisible. Answer: corrected as suggested.
Round 2
Reviewer 1 Report
The author has successfully addressed almost all my comments. I understand that a figure including all the images of the time series after co-registration could be too large for visualization. But I still would recommend the author to zoom in to a small section and show the registration in the small area (maybe the border between two images). Compared to the connectivity graph and numeric tables, such a figure could be more informative for the visual inspection of the registration accuracy.
Author Response
We have included a new figure (now Figure 6) to show three patches from the images before and after the alignment with the proposed MIRA procedure,
We would like to thank again Reviewer 1 for his/her comments.
Reviewer 2 Report
Reviewer does not have additional comments.
Author Response
The authors would like to thank again Reviewer 2 for the useful suggestions.
Reviewer 3 Report
Maybe I will not make other comments, I know that too many changes can make the paper worse, not better. Anyway, paper interesting and I hope it will be usefull for readers.
Author Response
The authors would like to thank again Reviewer 3 for the useful suggestions and his/her nice final comment.